# Chemistry, Biosynthesis and Pharmacology of Viniferin: Potential Resveratrol-Derived Molecules for New Drug Discovery, Development and Therapy

**DOI:** 10.3390/molecules27165072

**Published:** 2022-08-09

**Authors:** Shivkanya Fuloria, Mahendran Sekar, Farrah Syazana Khattulanuar, Siew Hua Gan, Nur Najihah Izzati Mat Rani, Subban Ravi, Vetriselvan Subramaniyan, Srikanth Jeyabalan, M. Yasmin Begum, Kumarappan Chidambaram, Kathiresan V. Sathasivam, Sher Zaman Safi, Yuan Seng Wu, Rusli Nordin, Mohammad Nazmul Hasan Maziz, Vinoth Kumarasamy, Pei Teng Lum, Neeraj Kumar Fuloria

**Affiliations:** 1Faculty of Pharmacy, AIMST University, Bedong 08100, Kedah, Malaysia; 2Department of Pharmaceutical Chemistry, Faculty of Pharmacy and Health Sciences, Royal College of Medicine Perak, Universiti Kuala Lumpur, Ipoh 30450, Perak, Malaysia; 3School of Pharmacy, Monash University Malaysia, Bandar Sunway 47500, Selangor, Malaysia; 4Faculty of Pharmacy and Health Sciences, Royal College of Medicine Perak, Universiti Kuala Lumpur, Ipoh 30450, Perak, Malaysia; 5Department of Chemistry, Karpagam Academy of Higher Education, Coimbatore 641021, India; 6Faculty of Medicine, Bioscience and Nursing, MAHSA University, Jalan SP 2, Bandar Saujana Putra, Jenjarom 42610, Selangor, Malaysia; 7Department of Pharmacology, Sri Ramachandra Faculty of Pharmacy, Sri Ramachandra Institute of Higher Education and Research (DU), Porur, Chennai 600116, India; 8Department of Pharmaceutics, College of Pharmacy, King Khalid University, Abha 61421, Saudi Arabia; 9Department of Pharmacology, College of Pharmacy, King Khalid University, Abha 62529, Saudi Arabia; 10Faculty of Applied Sciences, AIMST University, Bedong 08100, Kedah, Malaysia; 11Department of Biological Sciences and Centre for Virus and Vaccine Research, School of Medical and Life Sciences, Sunway University, Subang Jaya 47500, Selangor, Malaysia; 12Department of Preclinical Sciences, Faculty of Medicine and Health Sciences, Universiti Tunku Abdul Rahman, Kajang 43000, Selangor, Malaysia; 13Center for Transdisciplinary Research, Department of Pharmacology, Saveetha Dental College and Hospital, Saveetha Institute of Medical and Technical Sciences, Saveetha University, Chennai 600077, India

**Keywords:** viniferin, oligostilbenoid, chemistry, biosynthesis, pharmacology, drug discovery

## Abstract

Viniferin is a resveratrol derivative. Resveratrol is the most prominent stilbenoid synthesized by plants as a defense mechanism in response to microbial attack, toxins, infections or UV radiation. Different forms of viniferin exist, including alpha-viniferin (*α*-viniferin), beta-viniferin (β-viniferin), delta-viniferin (δ-viniferin), epsilon-viniferin (ε-viniferin), gamma-viniferin (γ-viniferin), R-viniferin (vitisin A), and R2-viniferin (vitisin B). All of these forms exhibit a range of important biological activities and, therefore, have several possible applications in clinical research and future drug development. In this review, we present a comprehensive literature search on the chemistry and biosynthesis of and the diverse studies conducted on viniferin, especially with regards to its anti-inflammatory, antipsoriasis, antidiabetic, antiplasmodic, anticancer, anti-angiogenic, antioxidant, anti-melanogenic, neurodegenerative effects, antiviral, antimicrobial, antifungal, antidiarrhea, anti-obesity and anthelminthic activities. In addition to highlighting its important chemical and biological activities, coherent and environmentally acceptable methods for establishing vinferin on a large scale are highlighted to allow the development of further research that can help to exploit its properties and develop new phyto-pharmaceuticals. Overall, viniferin and its derivatives have the potential to be the most effective nutritional supplement and supplementary medication, especially as a therapeutic approach. More researchers will be aware of viniferin as a pharmaceutical drug as a consequence of this review, and they will be encouraged to investigate viniferin and its derivatives as pharmaceutical drugs to prevent future health catastrophes caused by a variety of serious illnesses.

## 1. Introduction

Stilbenoids, also known as phytoalexins, are plant phenolics that are synthesized as a defense mechanism in response to abiotic and biotic stresses, such as microbial attack, toxins, infections or UV radiation [1,2]. The best known sources of stilbenoids are from *Vitis vinifera* [3]. Based on their structural characteristics, stilbenoids containing C6-C2-C6 backbone structures are further divided into five categories: stilbenes, oligostilbenes, bibenzyls, bisbibenzyls and phenanthrenes [1,2]. Among the stilbenoids, resveratrol is the most prominent and most investigated [1,4], while viniferins are more known as “resveratrol derivative” [5].

Several different forms of viniferin exist (Figure 1). The *α*-viniferin form is an oligostilbene of trimeric resveratrol [6] and was first found in *Caraganachamlagu Lam* as a compound that exhibits anti-inflammatory activities [7]. Furthermore, one of the major products of resveratrol-derived dehydrodimers is called *δ*-viniferin [8,9]. The δ-viniferin form is an isomer of ε-viniferin [10], which is also a dimer of resveratrol, extracted from *Vitis vinifera*; it has been extensively investigated for its potential benefits for human health [11,12,13]. Other oligomer stilbenoids that can be extracted and are found in the roots of *Vitis vinifera* are the resveratrol tetramers, R2-viniferin (Vitisin A) and R-viniferin (Vitisin B), which may mediate some other important biological activities [14].

To complete this review, relevant research was collected from several scientific databases, including Google Scholar, Scopus and PubMed. The literature search was performed using keywords such as “viniferin” AND “stillbenoid oligomers” OR “Vitis vinifera” AND “in-vitro” OR “in-vivo” OR “Biological studies” OR “Pharmacological studies” OR “Chemistry” OR “Toxicity studies” OR “Pharmacokinetics” for studies that had been published up until the date of the search. Studies that were not written in the English language or did not have abstracts were excluded. After applying the inclusion and exclusion criteria, as well as eliminating duplicates between the databases, a total of 73 studies were selected. The studies, classified into two major categories, phytochemistry and pharmacology, were further categorized based on the key findings, with no restriction on the dose, route, duration of administration, or type of study (animal or human). After a complete screening, the obtained information was summarized and included.

The studies indicated that viniferin is a potentially active molecule and that structural modifications to viniferin may lead to new drug development, with improved bioavailability and pharmacological action. The objective of this review is to discuss the chemistry and pharmacology of viniferin in order to examine how its derivatives might be useful molecules in the discovery of novel drugs to treat a variety of disorders.

## 2. Chemistry

### 2.1. Sources and Distribution of Viniferin

Viniferin is found in many plant species, among which grapes (Vitis vinifera) is a primary source. Table 1 summarizes details about the sources and distribution of various types of vinferin from medicinal plants.

### 2.2. Structural Characterization of Viniferin

A large number of resveratrol derivatives of higher structural complexity exist compared to simple substituted resveratrol analogues. The most common compounds found in nature are a variety of resveratrol dimers, such as ε-viniferin (**A**), δ-viniferin (**B**), and trimers, i.e., *α*-viniferin (**C**) (Figure 2).

Nevertheless, several important factors relating to stilbenoids’ nomenclature and structure may complicate its identification and classification. Another potential confusion lies in the structure and naming of viniferins. For example, although the compound itself takes the form of simple resveratrol dimers and trimers, there are two stereochemical centers at positions 7a and 8a on the dihydrofuran ring, allowing the formation of four potential stereoisomers. The *trans*- configuration of the two hydrogens in the saturated ring system provides alpha and beta hydrogen, unlike in the *cis* configuration, in which both hydrogens are either on the alpha side or in the beta position. The naming of the *cis* and *trans* conformations for these hydrogens is an area of confusion when dealing with viniferins, which also contain a *trans* (E) or *cis* (Z) double bond.

Determining the absolute configuration to differentiate between (+) and (−)-ε-viniferin, however, is more challenging. For known compounds, their [*α*] D values can be compared with those in the literature. Nevertheless, due to the difficulties in assigning absolute configurations to stilbenoid oligomers, many compounds have been reported with only their relative configurations assigned. Since (+)-ε-viniferin is considered a major stilbenoid intermediate for larger oligomers, the structures containing viniferin moieties are normally presented as containing the same configuration when not otherwise determined. In the many reports of complex oligomers, only the relative configuration is assigned [62].

#### *trans*-ε-Viniferin

The UV spectra in methanol (MeOH) showed λmax (log ε) values at 203 (5.05), 230 (4.87) and 324 nm (4.57), while in MeOH and sodium hydroxide (NaOH), they indicated λmax (log ε) at 211 (5.52), 244 (5.06) and 347 nm (4.84) [63,64]. The infra-red (IR) spectral data exhibited characteristic bands at 3393 cm^−1^ (OH), 1606, 1513, 1443 cm^−1^ (C=C aromatic) and 832 cm^−1^ (para-disubstituted benzene). The ^1^H-NMR spectra [63,64] were recorded in deuterated acetone with pairs of doublets appearing at δ 7.21 (2H, d, *J* = 9.0 Hz, H-2A and H-6A) and δ 6.83 (2H, d, *J* = 8.0 Hz, H-3A and H-5A), integrating two protons. They were assigned to the protons present in the aromatic ring A. The strong singlet at δ6.24 (3H, s, H-2B, 4B, 6B) for three protons was attributed to the protons present on ring B. The pair of doublets at δ 5.42 (1H, d, *J* = 5.0 Hz, H-1C) and 4.49 (1H, d, *J* = 5.0 Hz, H-2C) were due to the protons on ring C, while the signal at δ 6.32 (1H, d, *J* = 1.7 Hz, H-4D) was due to the meta-coupled proton H-4 on ring D. The H-2 proton of ring D appeared at δ 6.71, along with the protons of H-3E and H-5E of ring E. The signal at δ 7.18 (2H, d, *J* = 9.0 Hz, H-2E and H-6E) was attributed to the presence of the H-2E and H-6E protons on ring E. The alkene protons lying in between the two aromatic rings, D and E, appeared at δ 7.00 (1H, d, *J* = 15.1 Hz, H-â) and as a partially overlapped signal at δ 6.71.

The structural elucidation of *α*-viniferin is discussed in greater detail by Kitanaka et al. [65]. For example, *α*-viniferin with a molecular formula (C_42_H_30_O_9_) showed a peak for its pseudomolecular ion at *m*/*z* 701 [M + Na]^+^ and at *m*/*z* 678 for [M^+^] ion in its field-desorption–mass-spectrometry (FD-MS). In the UV spectra, the λmax peak appeared at 285 nm and in its IR spectra absorption bands at 3400 cm^−1^ for -OH group and at 1613 cm^−1^ for C=C, which are characteristic bands for the polyphenols observed.

In its ^13^C-NMR spectrum [65], *α*-viniferin exhibited forty-two signals, out of which six methine aliphatic carbon signals appeared between δ 46.4 and 95.6, twelve aromatic –CH groups appeared between δ 96.9 and 128.66, and a total of eighteen quaternary aromatic carbon atoms appeared between δ 118.0 and 161.7, including nine signals assigned to quaternary aromatic carbons under oxygen functions. The ^1^H-NMR spectrum exhibited three pairs of doublets for vicinally coupled methine protons at δ 3.97 (H_a_) and 6.07 (H_g_), 4.61 (H_b_, *J* = 6.4 Hz), 4.90 (H_d_, *J* = 6.4 Hz), 4.71 (H_c_, *J* = 9.7 Hz) and 5.95 (H_e_, *J* = 9.7 Hz). In addition, it also exhibited signals characteristic of three 1,2,3,5-tetrasubstituted benzene rings and three 1,4-disubstituted benzene rings.

The ^1^H-^1^H-COSY NMR spectrum confirmed the relationship between the three methine signals at δ 3.97 (H_a_), 4.71 (H_c_), 4.61 (H_b_), as well as the six meta-coupled signals at δ 5.99 (H_f_), 6.22 (H_h_), 6.72 (H_m_), 6.25 (H_j_), 6.59 (H_k_) and 6.23 (H_i_). There were cross peaks between the three signals of methine with the attached oxygen seen at 6.77 (H_g_), 5.95 (H_e_), 4.90 H_d_) and the six 4-hydroxy phenyl proton signals at δ 7.03 (H_p_), 6.72 (H_j_), 7.22 (H_r_), 6.77 (H_n_), 7.08 (H_q_) and 6.79 (H_o_). The plane structure of the *α*–viniferin was a ring structure with three 2-phenyl-2,3-dihydrobenzofuran units (I, II and III). All the proton signals werre assigned to the three units according to the coupling, beginning with the resonance of H_a_ (H-3 in Unit I) (Table 2). The assignments of all the methine and quaternary carbon signals were performed based on the ^13^C-^1^H- heteronuclear shift correlation spectrum.

The stereochemical configuration of the *α*–viniferin was determined based on the 2D nuclear Overhauser effect correlation spectroscopy (NOESY) spectrum. The appearance of cross peaks due to H_a_ (δ 3.97), H_p_ (δ 7.03), H_c_ (δ 4.71) and H_r_ (δ 7.22) indicated that the configurations at H-2 and H-3 n units I and II are *trans* to each other. The two cross peaks between H_a_ (δ 3.97), H_d_ (δ 4.90), H_c_ (δ 4.71) and H_b_ (δ 4.61) indicated that they exist on the same side in the plane structure. Consequently, the configuration at H_b_ and H_d_ is *trans.* The signals of H_g_ (δ 6.07) and H_e_ (δ 5.95) appeared at low field and were therefore deemed to be located at horizontal positions with respect to the aromatic ring A in units I and III, respectively. The proton signal of H_a_ (δ 3.97) appeared at a higher field than those of H_c_ (δ 4.71) and H_b_ (δ 4.61) because the atom lies above the plane of the aromatic ring in unit III.

The NMR data of δ-viniferin were reported by Teng et al. [66]. The ^1^H-NMR (500 MHz, CD_3_OD) d 7.56 (d, *J* = 1.6 Hz, 1H), 7.54 (d, *J* = 8.8 Hz, 2H), 7.52–7.49 (m, 1H), 7.48 (d, *J* = 8.5 Hz, 1H), 7.13 (d, *J* = 16.2 Hz, 1H), 6.95 (d, *J* = 16.2 Hz, 1H), 6.77 (d, *J* = 8.7 Hz, 2H), 6.49 (d, *J* = 2.2 Hz, 2H), 6.43 (d, *J* = 2.2 Hz, 2H), 6.35 (t, *J* = 2.2 Hz, 1H), 6.18 (t, *J* = 2.2 Hz, 1H). ^13^C-NMR (125 MHz, CD_3_OD) d 160.3 (2 C), 159.7 (2 C), 159.6, 154.8, 152.9, 141.0, 136.0, 134.1, 132.1, 129.8, 129.7 (2 C), 128.8, 123.9, 123.0, 118.6, 116.8, 116.4 (2 C), 111.8, 109.2 (2 C), 106.0 (2 C), 103.0 (2 C). HRESIMS: *m*/*z* 453.1336 [M + H]^+^ (calculated for C_28_H_21_O_6_, 453.13).

### 2.3. Biosynthesis

The biochemical synthesis of reverastrol in plants has been elucidated and occurs via a series of enzymatic processes, as highlighted in Figure 1. The synthesis begins with the amino acid phenylalanine, which is transformed into cinnamic acid and occurs by deamination; it is catalyzed by the enzyme phenylalanine ammonia lyase. The enzymatic hydroxylation to p-coumaric acid followed by the conversion of free acid into p-coumaroyl CoA occurs with the aid of CoA ligase. The final step in the synthesis involves the condensation of p-coumaroyl CoA (%) with malonyl CoA in the presence of stilbene synthase to furnish *trans*-resveratrol. Largely, resveratrol biosynthesis is controlled by stilbene synthase (STS), which controls the entry point into the stilbene biosynthetic pathway (Figure 1).

It was hypothesized that in nature, oligomerization proceeds via the formation of phenoxyl radical intermediates. Resveratrol oligomerization appears to proceed via the coupling of oxidatively generated phenoxyl radicals, as originally proposed by Langcake and Pryce [67]. The dimerization typically occurs (Figure 2) through two region-isomeric modes: the 8–10′ coupling (as found in ε-viniferin) and the 3–8′ coupling (δ-viniferin).

### 2.4. Bioavailability and Pharmacokinetics of Viniferin

Courtois et al. [68] reported that ε-viniferin is rich in carbons and hydrogens, which means that a) it is extremely poorly soluble in water, b) it has low bioavailability, and c) it easily undergoes isomerization under the influence of UV radiation. Nevertheless, by encapsulating the compound in phospholipid-based multi-lamellar liposomes (MLLs) called spherulites or onions, the photosensitivity is improved and the water solubility significantly increased [68]. In humans, ε-viniferin is mostly converted to glucuronides, and less often, to sulfates, whereas glucuronidation is the main pathway involved in rats [68]. The compound is rapidly glucuronidated by hepatic clearance [69], which explains its low bioavailability. In a study in 2018, it was reported that ε-viniferin accumulated in white adipose tissue, suggesting that these tissues may act as a reservoir for the native form, allowing slow release and long-term presence in the organism. Furthermore, ε-viniferin and its metabolite were found in higher concentrations in feces than in urine, signifying the main elimination pathway [70]. In addition, another form of viniferin, δ-viniferin, has a low bioavailability due to its low absorption and extensive metabolism, especially following oral administration compared to intravenous injection. The main metabolite found was glucuronide, followed by sulfates. It was further revealed that unmodified δ-viniferin and its metabolites were eliminated rapidly after intravenous injection and that δ-viniferin is primarily excreted unchanged in the feces after oral administration, with most appearing to be unabsorbed, according to the drug’s concentration in plasma [71]. A resveratrol trimer, *α*-viniferin is rapidly absorbed into the circulation and slowly eliminated, with only 4.2% bioavailability, following oral administration [7].

All the above-mentioned results indicate that viniferin has the potential to become a drug molecule for enhancing life span by potentially delaying ageing and preventing chronic illnesses. However, the limited bioavailability of viniferin is a major problem for converting these findings from fundamental research into clinical utility as a drug. Viniferin can potentially be made highly bioavailable through consumed with various foods, combination with other phytochemicals, the use of controlled-release technology, and the development of formulations using nanotechnology.

### 2.5. Medicinal Uses of Plants Containing Viniferin

*Paeonia suffruticosa* is an important traditional Chinese herb used to treat osteoarthritis (OA); oligostilbenes are the main active ingredients of its seeds [38]. Another plant, *vitis heyneana*, which is widely distributed in northern Vietnam, has been used in Vietnamese traditional medicine as an agent against arthritis, bronchitis, carbuncles, inflammatory conditions, and menstrual irregularities [35]. The dipterocarpaceae plant, *Cotylelobium melanoxylon*, which is widely distributed in Southeast Asia, has been used as an astringent, antilaxative and blood-coagulation agent in traditional Thai medicine [72]. *Dipterocarpus littoralis*, commonly known as Meranti Jawa in Indonesia, is traditionally used to treat diseases such as diarrhea, diabetes and malaria [27] (Figure 3). Another important plant is *Shorea roxburghii* (Dipterocarpaceae), which is widely distributed in Thailand and its neighboring countries, such as Cambodia, India, Laos, Malaysia, Myanmar and Vietnam. The bark of *Shorea roxburghii* (“Phayom” in Thailand) has been used as an astringent or a preservative in traditional beverages in Thailand [73]. In Indian folk medicine, the plant has been used in the treatment of dysentery, diarrhea and cholera [46].

The genus Hopea (Dipterocarpaceae), which consists of over 104 species, is distributed primarily in southern parts of India and China and in Sri Lanka. *Hopea ponga* (Dennst.) Mabb is an endemic tree found mainly in the tropical evergreen forests of the South Western Ghats of India. The plant was reportedly used in traditional medicine in the treatment of diabetes, piles and snake bites [32]. Another plant, *Vitis amurensis* Rupr. (Vitaceae) is a wild-growing grape, widely distributed in Korea, China and Japan. Its fruit has been used as a raw material for juice and wine in three different countries. The root and stem have been used to relieve pain from injury, rheumatalgia, stomach ache, neuralgic pain and abdominal pain [74].

*Caragana sinica* (Buchoz) Rehd. (Fabaceae), a deciduous shrub, is widely distributed in Korea, China and Japan. Its dried roots have been used in the treatment of asthenia syndrome, vascular hypertension, leukorrhagia, bruises, contusion, rheumatism, neuralgia, arthritis and migraine as a folk medicine [19]. The underground parts of *C. chamlague*, which have been used in Korea and China as folk medicine, are purported to be effective against neuralgia, rheumatism and arthritis [75].

## 3. Biological Properties of Viniferin

### 3.1. Anti-Inflammatory Effects

A study by Vion et al. [76] reported that *trans* ε-viniferin decreased the amount of inflammatory mediators, such as TNF*α* and IL-6. In another study on knee damage associated with arthritis, it was reported that the active constituents of *Vitis thunbergii* var. taiwaniana, including resveratrol, hopeaphenol and (+)-ε-viniferin, significantly scavenged 2,2-diphenyl-1-picrylhydrazyl (DPPH) radicals and inhibited prostaglandin E2 (PGE2) production in lipopolysaccharide (LPS)-induced penehyclidine hydrochloride (PHC)s. Additionally, there was a significant decrease in serum PGE2 and 2-18F-fluoro-2-deoxy-D-glucose (18F-FDG) levels in LPS-induced acute inflammatory arthritis in rabbits [53]. In a recent study, ten oligostilbenes extracted from the seed of *Paeonia suffruticosa* showed protective effects at low concentrations on osteoarthritis chondrocytes. One of the compounds is *trans*-viniferin, which tends to be most effective in promoting the expressions of Collagen Type II Alpha 1 Chain (COL2A1) and SRY-Box Transcription Factor 9 (SOX9) [38]. On the other hand, the oral and IV administration of *α*-Viniferin at >30 mg/kg and >3 mg/kg, respectively, showed significant anti-inflammatory effects on carrageenin-induced paw edema in mice. The compound also showed an inhibitory effect on COX-2 activity and a very weak inhibitory effect on COX-1 activity [24]. These findings are supported by the report by Chung et al. [77], who investigated the anti-inflammatory effects of *α*-viniferin and established that it inhibits ERK-mediated STAT-1 activation in IFN-γ–stimulated macrophages, thus downregulating STAT-1-inducible inflammatory genes [77].

Among the many oligostilbenoids extracted from *Vitis heyneana,*
*α*-viniferin has the highest potential inhibitory activities. Overall, LPS-induced COX-2 expression and PGE2 production were suppressed, nitric oxide (NO) release was significantly reduced in a dose-dependent manner and the activation of the transcription factor of NF-κB was inhibited [47]. A study on *Vitis vinifera* root extract, including seven stillbenoids (resveratrol, piceatannol, *trans*-*ε*-viniferin, ampelopsin-A, miyabenol C, R-2-viniferin (Vitisin A) and R-viniferin (Vitisin B)) established that the extract has potent free-radical-scavenging activity in terms of DPPH, hydroxyl and galvinoxyl, in a dose-dependent manner; the superoxide radicals are also scavenged when the extract is used in high concentrations. Additionally, it protects against DNA damage caused by hydrogen peroxide while downregulating pro-inflammatory gene expression, including IL-1β and iNOS in cultured macrophages [59].

### 3.2. Antidiabetic Effects

An earlier study established that the methanolic extracts from the wood and bark of *Cotylelobium melanoxylon* could inhibit elevations in plasma glucose following sucrose loading in rats and ameliorates triglyceride elevation following olive-oil loading in mice. In the study, *cis*- (+)-ε-viniferin was isolated from the bark extract, while (+)-ε-viniferin was isolated from both wood and bark extracts [72]. The ε-viniferin caused a significant reduction in the concentrations of fasting blood glucose (FBG), total cholesterol (TC), triglyceride (TG) and low-density-lipoprotein cholesterol (LDL-C). Additionally, Liu et al. [78] found that the glucose-tolerance and liver- and kidney-damage indices, such as alanine aminotransferase (ALT), aspartate aminotransaminase (AST), creatinine (CR) and blood urea nitrogen (BUN) of diabetic rats also improved. Furthermore, the activation of AMP-activated protein kinase (AMPK) was also increased and the histopathological changes were attenuated in the livers of diabetic rats by binding to the hinge region between the *α*- and β-units of AMPK, as well as interacting with the active site of AMPK [78].

The progression of diabetes mellitus (DM) can be ameliorated by inhibiting *α*-glucosidase, which delays glucose absorption and lowers postprandial blood-glucose levels. Lulan et al. [27], who extracted *α*-viniferin from *Dipterocarpus littoralis*, established its antidiabetic potential, which acts by inhibiting the activities of the *α*-glucosidase and *α*-amylase of rats in the intestine. The study compared the extract’s activity with that of acarbose as a standard. The finding was also supported by an earlier study by Morikawa et al. [46], who reported that oral (+)-*α*-viniferin showed an inhibitory activity against plasma glucose elevation in sucrose-loaded rats at 100–200 mg/kg through the inhibition of intestinal *α*-glucosidase and aldose reductase activities [46].

In another study, for the first time, acetone and ethanol extracts from the stem bark of *Hopea ponga* (Dennst.) Mabb were tested for their antidiabetic activity. Both (−)-*ε*-viniferin and (−)-*α*-viniferin, which were among the ten compounds isolated, showed inhibition towards the activities of *α*-glucosidase and *α*-amylase, with prominent antiglycation activity seen. It was also observed that *α*-viniferin can increase glucose uptake, mainly due to AMPK upregulation, which eventually leads to the translocation of the glucose transporter (GLUT-4) into the cell membrane [32]. A recent study by Oranje et al. [79] investigated sodium-glucose co-transporter 1 (SLGT 1) and 2 (SLGT 2), which are targets for glycemic control in type 2 diabetes mellitus. They established that the isomers of the resveratrol dimers (+)-ε-viniferin and (−)-ε-viniferin inhibit the sodium-glucose co-transporter, while (+)-ε-viniferin inhibits SLGT 1 by 44%, with little inhibition shown towards SLGT2. Nevertheless, by contrast, (−)-ε-viniferin did not inhibit SGLT1, but did show a 35% inhibition of SGLT2. Another study on racemic forms of *trans*-*δ*-viniferin and *trans*-*ε*-viniferin also found that both had higher efficacy in inhibiting pancreatic alpha-amylase compared to pure enantiomers [9].

### 3.3. Anticancer Effects

Most studies on resveratrol oligomers, including viniferin, are focused on its anticancer activity. It has been reported that a combination of a first-generation platinum complex, the anti-cancer drug cisplatin (CDDP), and ε-viniferin, a natural antioxidant, has strong apoptotic effects on the glioma cell lines (C6) when used in low concentrations, compared to using the compound alone [80]. Previously, researchers have reported that the apoptosis of hepatocellular carcinoma (HepG2) cells may be induced by using a combination of vincristine and ε-viniferine [81]. Their finding was supported by another study, which investigated the anticancer activity of the combination of vincristine and ε-viniferine loaded with PLGA-b-PEG nanoparticles and also established that the combination induces apoptosis in HepG2 cells [80].

Another study was conducted on the anticancer activity in human hepatocellular carcinoma (HCC) cell lines p53 wild-type HepG2 and p53-null Hep3B. R2-viniferin inhibited HepG2 but not Hep3B, arrested the cell cycle at G2/M and increased the intracellular reactive oxygen species (ROS), caspase 3 activity and the ratio of Bax/Bcl-2 proteins, indicative of apoptosis [82] (Figure 4). R2-viniferin was also tested on the canine glioblastoma cell line D-GBM and the canine histiocytic sarcoma cell line DH82. The author used Vineatrol^®^30, which contains resveratrol and its oligomers (R2-viniferin and hopeaphenol) which were confirmed to exert a potent anti-proliferative effect on the two canine tumor-cell lines. The effect, at least in D-GBM cells, is due to the induction of apoptosis via the activation of caspase 9 and 3/7 [83]. Subsequently, the researchers performed a comparison of the anticancer activity of R-viniferin and resveratrol against the prostate cancer cell line lymph node carcinoma of the prostate (LNCaP). They established that both compounds can inhibit cell growth and arrest the G1 phase cell cycle, although R-viniferin was more potent and tended to increase the apoptotic cellular fraction, along with increasing the activity of apoptosis-associated enzymes [14].

Additionally, *α*-viniferin was also reported to be effective against colon cancer cell lines (HCT-116, HT-29, Caco-2) by blocking the S-phase of the cell cycle. Nevertheless, no apoptotic effect was induced [84]. Additionally, *α*-viniferin has antiproliferative effects against chronic myelogenous leukemia (CML). In vitro, the said compound, along with resveratrol, significantly inhibited the proliferation of K562 cells in both dose- and time-dependent manners by reducing the expression of the BCR-ABL protein. A high dose of *α*-viniferin caused serious cell death, cell fragmentation, and nuclei lysis, indicating apoptosis [85].

A study on the anticancer activity of *α*-viniferin against human prostate cancer (PCa) cells reported that it has antiproliferative effects on LNCaP, DU145 and PC-3 cancer cells, depending on the dose and timing of treatment, while conferring strong cytotoxicity in non-androgen-dependent PCa cells. The compound inhibited AR downstream expression in LNCaP cells and inhibited the activation of the GR signaling pathway in the DU145 and PC-3 cell lines. Additionally, it also induced cancer cell apoptosis through the AMPK-mediated activation of autophagy and inhibited the expression of the glucocorticoid receptor (GR) in castration-resistant prostate cancer (CRPC) [86]. In terms of testing the anticancer activity on human melanoma cells, ε-viniferin blocks the cell cycle of melanoma cells in the S-phase by modulating the key regulators of the cell cycle, i.e., cyclins A, E, D1 and their cyclin–dependent kinases 1 and 2, which are associated with the induction of cell death, including apoptosis and necrosis [87].

On the other hand, a study [29] discovered that (+)-*α*-viniferin and resveratrol possessed antiproliferative action against SK-MEL-28 melanoma cells, where (+)-*α*-viniferin was reported to be more potent. The compound arrests the G1 cell cycle, as well as inducing DNA damage followed by the induction of apoptosis in SK-MEL-28 cells, which was confirmed by an increased expression of *γ*-H2AX and cleaved caspase-3. Additionally, (+)-*α*-viniferin and resveratrol significantly decreased the expression of cyclin B1, which is important for G2/M phase transition in the cell cycle.

### 3.4. Anti-Angiogenic Effects

Atherosclerosis can be prevented by protecting the vascular endothelial cells (VECs). In fact, low concentrations of ε- and δ-viniferin significantly stimulate wound repair via nitric oxide (NO) production, the activation of endothelial NO synthase and the induction of sirtuin 1 (SIRT1) and HO-1 expression [88]. These findings were supported by another study, which confirmed the inhibition of vascular arginase activity involved in the production of NO by ε-viniferin [89]. The anti-angiogenic effects of *α*-viniferin were observed when it inhibited mitogen-induced human-umbilical-vein endothelial cell (HUVEC) proliferation through the hypophosphorylation of retinoblastoma protein. It also suppressed mitogen-induced HUVEC adhesion, migration, invasion, and microvessel outgrowth, as mediated by the downregulation of cell-cycle-related proteins, vascular endothelial growth factor receptor-2 (VEGFR-2), and matrix metalloproteinase-2. The inactivation of the VEGFR-2/p70 ribosomal S6 kinase signaling pathway was involved in the *α*-viniferin-mediated modulation of endothelial cell responses [20].

In addition, (+)-vitisin A can effectively reduce 24-hour systolic and diastolic blood pressures following a single oral dose administered at spontaneously hypertensive rats (SHRs). It also exhibits anti-angiotensin-converting enzyme (ACE-I) and vasodilating effects against phenylephrine-induced tensions in an endothelium-intact aortic ring of the SHRs [90]. In addition to (+)-vitisin A, it was confirmed that ε-viniferin possessed similar activity. The compound induces the proliferation and wound repair in VECs via NO production and is involved in the protection of VECs from oxidative-stress-induced cell death. It also inhibited tACE activity in vitro and eventually reduced blood pressure to improve the cardiac mass in SHRs [91].

### 3.5. Anti-Melanogenic Effects

Facial hyperpigmentation was reported to be improved following the skin application of topical products containing *α*-viniferin on the skin. It has been reported that *α*-viniferin inhibited melanin production in *α*-melanocyte-stimulating hormone (*α*-MSH)-, histamine- or cell-permeable cAMP-activated melanocyte cultures. It also decreased the melanin index on facial melasma and freckles in humans. The *α*-viniferin accelerated protein kinase A (PKA) inactivation via the reassociation between catalytic and regulatory subunits in cAMP-elevated melanocytes, a feedback loop in the melanogenic process. Consequently, the cAMP/PKA-signaled phosphorylation of cAMP-responsive element-binding protein (CREB) coupled with the dephosphorylation of cAMP-regulated transcriptional co-activator 1 (CRTC1), which was inhibited; hence, the expression of the MITF-M or Tyro gene was downregulated with decreased melanin pigmentation [92].

### 3.6. Anti-Obesity Effects

A study compared both the in vitro and the in vivo anti-obesity effect of ε-viniferin and t-resveratrol. The ε-viniferin was confirmed to have a higher anti-adipogenesis activity in 3T3-L1 cells. It significantly suppressed lipid accumulation and the expression of the adipogenesis marker gene, PPAR gamma. When compared with a high-fat-diet control mice group, there was reduced body weight, as well as liver triglyceride levels following ε-viniferin treatment. In the meantime, the levels of plasma insulin and leptin were significantly improved [93]. The (+)-ε-viniferin extracted from the roots of *Vitis thunbergii* var. taiwaniana 1 (VTT-R) significantly reduces the lipid deposits in 3T3-L1 adipocytes and inhibits 3-hydroxy-3-methylglutaryl-CoA (HMG-CoA) reductase (Figure 5). The compound is believed to lower the body weights of mice, the weight ratio of mesenteric fat, blood glucose, total cholesterol, and low-density lipoprotein in high-fat-diet-induced obesity groups [52].

### 3.7. Antidiarrheal Effects

Yu et al. [93] reported that *trans*-*ε*-viniferin and R2-viniferin possess antisecretory effects and are useful in the treatment of diarrhea. Additionally, the compound inhibited the activation of intestinal calcium-activated chloride channel (CaCC) when tested on a neonatal mouse model of rotaviral diarrhea. It suppressed diarrhea without affecting the rotaviral infection. Furthermore, the *trans*-*ε*-viniferin inhibited the physiologically relevant, long-term CaCC current following agonist stimulation, without affecting cytoplasmic Ca^2+^ signaling, with both compounds believed to inhibit short-circuit currents in the mouse colon [93]. Subsequently, the author investigated the role of *trans*-*δ*-viniferin in rotavirus-infected diarrhea and inflammatory-bowel-syndrome diarrhea IBS-D. They found that the resveratrol dimer could inhibit TMEM16A activity in TMEM16A-expressed Fischer rat thyroid (FRT) cells, as well as preventing Ca^2+^-activated Cl− current in HT-29 cells and in the colonic mucosa. Moreover, the compound prevents diarrhea caused by rotaviral infection and reduces the pellet number in IBS-D mice [94].

### 3.8. Neuroprotective Effects

Alzheimer’s disease (AD) affects many cellular and molecular targets; therefore, its therapy requires multi-target molecules. Caillaud et al. [95] evaluated the effects of *trans*-*ε*-viniferin as a neuroprotective agent on transgenic APPswePS1dE9 mice. They reported that the compound can cross the blood–brain barrier and reduce the size, as well as the density, of amyloid deposits to ameliorate astrocyte and microglial reactivity. The effect was shown only after 3-to-6-month-old mice were intraperitoneally injected (10 mg/kg) every week [96]. Additionally, a study found that AD may be caused by the accumulation and aggregation of abnormal b-amyloid peptide and suggested that the inhibition of b-amyloid (Ab) fibril formation is helpful in treating AD. The researchers reported that *ε*-viniferin glucoside inhibits Ab (25–35) fibril formation in vitro. Additionally, the effects of *ε*-viniferin on the aggregation of the full-length peptides Ab (1–40) and Ab (1–42)] and on b-amyloid-induced toxicity was investigated in PC12 cells; ε-viniferin was confirmed to inhibit Ab cytotoxicity [96].

Furthermore, (+)-*α*-viniferin was also confirmed to be one of the most important natural constituents to exhibit anti-acetylcholinesterase (AChE) activity, being a significantly specific, reversible and non-competitive AChE inhibitor. Overall, AChE inhibitors increase the efficiency of cholinergic transmission by preventing the hydrolysis of released ACh, allowing more ACh to become available at the cholinergic synapse [75]. In addition, *α*-viniferin can prevent and treat AD by enhancing alpha-secretase ADAM10 gene expression. Consequently, it prevents the formation of toxic amyloid beta peptides, but also provides a neuroprotective fragment of the amyloid precursor protein (sAPPalpha). However, a challenge remains due to its limitation in crossing the blood–brain barrier [21], making the design of new formulations a necessity in the future.

In addition to AD, there is also an inherited neurodegenerative disorder known as the Huntington disease (HD). HD is an incurable disease occurring due to an abnormal polyglutamine expansion in the protein named Huntingtin. A study demonstrated that *trans*-(−)-*ε*-viniferin can increase the levels of mitochondrial Sirtuin 3 (SIRT3), activates AMPK and protects cells in models of HD [97]. Moreover, ε-viniferin can upregulate SIRT3 expression, which promotes FOXO3 deacetylation and nuclear localization as well as increasing ATP production and decreasing ROS production. The compound also maintains mitochondrial homeostasis, thus inhibiting rotenone-induced cell apoptosis, making ε-viniferin a potential treatment for neurodegenerative disorders [98].

### 3.9. Antioxidant Effects

A recent study demonstrated that scratched vascular endothelial cells (VECs) treated with resveratrol (10 μM), ε-viniferin (10 μM) and δ-viniferin (5 μM) significantly reversed decreased cell viability after the addition of hydrogen peroxide to the cells, indicating that these compounds are resistant to oxidative stress by increasing the catalase protein level [88]. In addition, ε-viniferin is a potent antioxidant when tested on muscadine grape (*Vitis rotundifolia*) hairy root cultures, acting via its radical scavenging capacity [50]. Furthermore, *α*-viniferin exhibited antioxidant activity in cupric ion-reducing antioxidant-capacity, ferric-reducing antioxidant-power, DPPH scavenging, and 2-phenyl- 4,4,5,5-tetramethylimidazoline-1-oxyl 3-oxide radical-scavenging assays. The author concluded that this involved redox-mediated mechanisms, especially electron and H+-transfer, as well as non-redox-mediated mechanisms, including Fe2+-chelation or radical adduct formation [99].

The dimer of resveratrol, *trans*-*δ*-viniferin, exhibited moderate antioxidant activity when tested using in vitro model systems, including hydroxy radical scavenging, DPPH and lipid peroxidation. Among the oxygen radicals, the hydroxyl radical is the most reactive and causes great damage to living cells due to its ability to react with various molecules, such as phospholipids, DNA and organic acids. The effects of this compound on human erythrocytes have also been confirmed to protect red blood cells from hemoglobin oxidation [100,101].

*trans*- and *cis*-ε-viniferins were among the stilbene derivatives isolated from the seeds of *Paeonia lactiflora.* The compounds were evaluated against the 2-deoxyribose degradation and rat-liver microsomal lipid peroxidation induced by the hydroxyl radical generated via a Fenton-type reaction. It was found that *trans*-*ε*-viniferin significantly inhibited the degradation of 2-deoxyribose and rat-liver microsomal lipid peroxidation, whereas *cis*-ε-viniferin exerted only moderate antioxidant activity [102].

### 3.10. Antiplasmodic Effects

Malaria is a life-threatening disease caused by parasite species that can infect humans. Among these, *Plasmodium falciparum* and *Plasmodium vivax* are the most dangerous. It has been reported that in 2020, nearly 50% of the world’s population was at risk of malaria [103]. To date, many attempts have been made to both prevent and treat malaria, one of which involved the use of medicinal herbs in traditional remedies. The World Health Organization (WHO) has recommended preventive strategies to combat the disease by using antimalarial drugs. Many studies are conducted to determine suitable compounds in plants as health interventions. It has been reported [27] that the bioactive compounds isolated from *Dipterocarpus littoralis*, especially *α*-viniferin, has good antiplasmodiac activity. Based on their comprehensive spectrum analyses, including IR, 1D, and 2D NMR, as well as comparisons with research data, the structure of *α*-viniferin (referred to as “Compound 1”) was determined. It showed alpha-glucosidase and alpha-amylase inhibitory activities with 50% inhibitory concentration (IC_50_) values of 256.17 and 212.79 μg/mL respectively. The antiplasmodial activity was tested in vitro against the plasmodium falciparum strain 3D7 at 100 g/mL and demonstrated substantial antiplasmodial inhibitory activity (IC_50_ value of 2.76 g/mL). Based on the findings, the isolated extract from *Dipterocarpus littoralis*, *α*-viniferin, is a potential source to be developed into an antiplasmodial agent [27] (Figure 6).

### 3.11. Antimicrobial Effects

According to the WHO, antimicrobial resistance (AMR) is one of the most significant threats to global public health. Antimicrobial resistance (AMR) develops when bacteria, viruses, fungi and parasites evolve over time and lose their ability to respond to antibiotics, making infections more difficult to treat, as well as raising the risk of disease transmission, severe illness, and death [104].

According to Schnee et al. [105], crude extracts of *Vitis vinifera* canes have considerable antifungal activities. The six identified compounds (ampelopsin A, hopeaphenol, *trans*-resveratrol, ampelopsin H, ε-viniferin, and E-vitisin B) are active against *Plasmopara viticola*, the pathogen considered the most damaging, affecting grapevines. Moreover, ε-viniferin exhibited low antifungal activity against *Botrytis cinerea*. Nevertheless, none of the identified compounds has been reported to inhibit the germination of *E. necator* [105].

The study conducted by Yadav et al. [106] indicated that viniferin compounds restrained *S. pneumoniae* growth and destroyed bacteria in biofilms. Viniferin treatment impairs the membrane integrity of biofilm bacteria, according to scanning electron microscopy (SEM) examination and live/dead biofilm staining of pre-established biofilms. Viniferin affects bacterial cell permeability and eventually kills bacteria, according to crystal violet absorption, total protein, and DNA and RNA release. Therefore, viniferin’s fatal action is purported to cause a change in cell-membrane permeability. Although viniferin is commonly reported to have anti-cancer and anti-obesity effects, the investigators focused on its unique antibacterial and antibiofilm against *S. pneumonia,* which make viniferin and its derivatives good candidates for the development of novel pneumococcal antimicrobial drugs [106].

Another study reported that the resveratrol dimer (dehydro-δ-viniferin), a natural stilbenoid with a benzofuran core, is a potential antimicrobial agent against Gram-positive bacteria, especially the foodborne pathogen, *Listeria monocytogenes* [107]. *Listeria monocytogenes* can infect both humans and animals, although it is difficult to control the pathogen due to its ability to build biofilms. The virus has been isolated from a wide range of foods, including raw milk, cheese, raw meat products and salads, making it extremely common in food production and distribution. This bactere can cause several diseases, mainly gastroenteritis, endocarditis, rhombencephalitis, invasive listeriosis, septicemia, meningitis and neonatal infections; it can also lead to abortion [108]. Mattio et al. [107] utilized various protocols to derive stilbenoid-derived 2,3-diaryl-5-substituted benzofurans and found that key stages, such as the demethylation of phenolic groups, are required. *Staphylococcus aureus (S. aureus)* ATCC29213 was used to test antibacterial activity and the results showed that 5,5′-(2-(4-hydroxyphenyl)benzofuran-3,5-diyl)bis(benzene-1,3-diol) analogue is an important potential compound for further investigation.

In addition, Rahim et al. [109] tested another form of viniferin, *α*-viniferin, as a potential antibacterial agent against *S. aureus*, a multidrug-resistant bacterium that is prone to serious healthcare-associated and community-acquired infections globally. The nasal-colonization bacterium can result in a variety of diseases, ranging from mild to life-threatening, including pneumonia, chronic osteomyelitis, and bacteremia. The aim of the study was to use culture-based procedures to explore the antibacterial efficiency of *α*-viniferin against the normal nares microflora, *S. aureus* and *methicillin resistance staphylococcus aureus* (MRSA). The experiment involved a ten-day clinical trial and indicated that *α*-viniferin demonstrated 50% minimum inhibitory concentrations (MIC50 values) of 7.8 g/mL in culture broth medium throughout the ten-day clinical experiment. A sterile cotton swab stick was used to deliver *α*-viniferin three times a day for ten days in the nares. The nasal-swab samples were collected at baseline and after 10 days and evaluated. The number of *S. aureus* was greatly reduced in the cultures, as further confirmed by the reverse transcriptase polymerase chain reaction (RT PCR)-based analysis (0.01). Furthermore, the 16S ribosomal RNA-based amplicon-sequencing study revealed a reduction from 23.99 to 51.03% in the *S. aureus* at the genus level. The findings showed that *α*-viniferin is an effective antibacterial agent against the Staphylococcus group, especially against *S. aureus* and MRSA, but showed no activity against other nasal microflora. Furthermore, *α*-viniferin enhanced skin moisture content to maintain skin plasticity and barrier integrity in the absence of toxicity. In conclusion, the research used a clinical trial to demonstrate the clinical effectiveness of viniferin as a possible candidate against *S. aureus* [109].

According to Mattivi et al. [110], viniferins are a small group of *trans*-resveratrol oligomers, detected in the Vitaceae family, with antifungal characteristics, thus allowing plants to resist attacks from pathogens. The study was performed by isolating and characterizing the entire class of viniferins accumulated in the leaves of hybrid *Vitis vinifera* genotypes infected by *Plasmopara viticola*. Six days after infection, the infected leaves of resistant plants were collected, extracted with methanol and pre-purified using ENV+ and Toyopearl HW 40S resins by flash chromatography.

Seven dimers (six stilbenes and one stilbenoid) were detected in infected leaves. Ampelopsin D, quadrangularin A, E-ϵ-viniferin and Z-viniferin were four compounds new to the grapevine. Next, four trimers (three stilbenes and one stilbenoid), two of which (Z-miyabenol C and E-*cis*-miyabenol C) were found that were new to the grapevine, as well as three tetramer stilbenoids, isohopeaphenol, ampelopsin H, all new to the grapevine, as well as a vaticanol C-like isomer. Other preformed phenolics are structurally changed in tissues infected with *P. viticola*, as evidenced by the isolation of a dimer derived from the condensation of (+)-catechin with *trans*-caffeic acid [110].

Ultra-high-performance liquid-chromatography–mass -spectrometry (UHPLC-MS) was used to evaluate stilbene-enriched extracts from the waste of *Vitis vinifera* (cane, wood, and root). Eleven stilbenes were identified (ampelopsin A, (E)-piceatannol, pallidol, (E)-resveratrol, hopeaphenol, isohopeaphenol, (E)-ϵ-viniferin, (E)-miyabenol C, (E)-ω-viniferin, R2-viniferin and r-viniferin) and quantified. The IC_50_ for *Plasmopara viticola* sporulation growth was calculated. The R-viniferin had the lowest IC_50_ (highest efficacy) against *Plasmopara viticola*, followed by hopeaphenol and R2-viniferin. The antifungal activity of the stilbene extracts was highest in the wood extract, followed by the root extract. Overall, the findings indicate that the four most active chemicals (R-viniferin, R2-viniferin, hopeaphenol, and isohopeaphenol) of the stilbene complex combinations derived from the *Vitis vinifera* waste found in both wood and roots can be exploited for the development of natural fungicides as a low-cost source of bioactive stilbenes [111].

### 3.12. Antihelmintic Effects

Viniferin has been investigated for its potential antihelminthic effects. Roy and Giri [22] reported that *α*-viniferin is an active compound found in *Carex baccans (C. baccans) L*., a plant known to have anti-diabetic, anti-inflammatory and anticancer activities. Different tribes in Northeast India have traditionally consumed *C. baccans* to treat intestinal worm infections. In in vitro tests, helminths were exposed to different amounts of *α*-viniferin (50, 100, and 200 M/mL in physiological buffered saline), followed by measurements of motility and mortality.

The activity of vital tegumental enzymes, such as acid phosphatase, alkaline phosphatase and adenosine triphosphatase, was reduced in parasites exposed to *α*-viniferin in histochemical and biochemical studies. The extensive structural and functional alterations observed in the treated parasites are indicative of the compound’s cestocidal activity. The deformation and destruction of suckers seen in resveratrol-exposed *Raillientina echinobothrida* (*R. echinobothrida)* add to the phytochemical’s anthelmintic potential. The NOS and AChE activities change in *R. echinobothrida* following exposure to resveratrol and *α*-viniferin imply that both phytochemicals have anthelmintic potential [112].

## 4. Industrial Application of Viniferin

Stilbenoids are a group of organic compounds with C6-C2-C6 as the structural formula. They are found in a range of plant species, including *Vitis vinifera*; as with those in grapes, they are naturally occurring. Resveratrol is the most prominent and frequently investigated stilbenoid [4], while viniferins are also known as resveratrol derivatives [5]. Some resveratrol derivatives, such as piceatannol, pterostilbene and ε-viniferin have recently piqued the interest of industries [113]. Stilbenes are a family of phenolic secondary metabolites known for their important roles in plant protection and human health [114]. The potential applications of viniferins in medical technology and pharmaceutical industries are essential to health, since resveratrol derivatives have a wide range of positive health effects (anti-inflammatory, antidiabetic, anticancer, antiangiogenic, antimelanogenic, anti-obesity, anti-diarrheal and antioxidant). For example, a substantial number of traditional Chinese medications (TCM) have been shown to contain stilbene *α*-viniferin and confer some effects on leukemic cells. According to the National Cancer Institute Developmental Therapeutics Program records (NSC 655524), leukemia and central-nervous-system cell lines are responsive to *α*-viniferin treatments in vitro.

In the agricultural industry, numerous studies have shown that vine shoots, one of the most abundant winery wastes, are useful sources of bioactive compounds, such as stilbenes. The predominant stilbenoids in vine shoots are *trans*-resveratrol and ε-viniferin, whose content varies depending on numerous intrinsic and extrinsic factors [114]. Since other sources of stilbenoids, such as peanuts, pistachios, peanut butter and chocolates, can also offer health benefits, consuming them in ideal quantities is recommended [58]. Given the potential and influence of stilbenoids, particularly on plant physiology, the agriculture sector is the most affected in terms of their usage. Overall, when contemplating the industrial applications of stilbenoids, for example, the antifungal properties of resveratrol in various leaves and berries are critical [115].

Viniferin is used for both its nutraceutical and its cosmeceutical effects. According to Malinowska et al. [116], grape canes are viticulture-waste biomasses that contain bioactive polyphenols that are useful in cosmetics. Although various studies have examined the cosmetic properties of E-resveratrol, only a few have investigated the potential of ε-viniferin, the second most abundant ingredient in grape cane extracts (GCE). GCE from polyphenol-rich grape types can be used as a multifunctional cosmetic component. The skin-whitening potential of GCE was compared to those of pure ε-resveratrol and ε-viniferin using a tyrosinase-inhibition assay and the activation capability of the cell-longevity SIRT1 protein of GCE. Overall, the current findings allowed the GCE from polyphenol-rich types to be considered as multifunctional cosmetic components, in compliance with green chemistry principles. For example, the *Vitis vinifera*-derived ingredients included in the safety assessment are reported to have many possible functions in cosmetic formulations. *Vitis Vinifera* (grape) seed extract is reported to function as an anti-caries, anti-dandruff, anti-fungal, anti-microbial, antioxidant, flavoring, light stabilizer, oral care, oral-hygiene and sunscreen agent. A panel that reviewed the safety of *Vitis vinifera*-derived components (n = 24) determined that their application is safe in current cosmetics use and concentrations. The chemicals are most commonly applied as skin conditioners in cosmetics. Antioxidants, flavoring agents, and/or colorants are confirmed to be present in some of these components. Additionally, certain grape compounds have been evaluated for safety as cosmetic additives in the past; others have not [117].

## 5. Structurally-Related Viniferin Molecules for New Drug Discovery and Development

Various nomenclature and structures exist in the literature, which complicate the identification and classification of stilbenoids, particularly viniferins. Viniferins are oligomers of resveratrol; however, there are two stereochemical centers on the dihydrofuran ring, allowing the formation of four potential stereoisomers. The naming of *cis* and *trans*-conformations for hydrogen is an area of confusion and is further complicated when stilbenoids also contain a *trans*- (E) or *cis*- (Z) double bond. Determining the absolute configuration (the difference between (+) and (−) viniferins) is more challenging. Due to the challenges in assigning absolute configurations to stilbenoid oligomers, many compounds have been reported, with only their relative configurations assigned thus far [62].

E-δ-viniferin **(1**) (E-resveratrol dehydrodimer) has been reported to be present in *V. vinifera* cell-suspension cultures, leaves and wine [115]. Its glycosides, E-δ–viniferin-11-O-β-D glucopyranoside **(1a)** (resveratrol dehydrodimer 11-O-β-D-glucopyranoside) and E-δ-Viniferin 11′-O-β-D-glucopyranoside (E-resveratrol dehydrodimer 11′-O-β-D-glucopyranoside) (**1b**), are reported from *V. vinifera* cell-suspension cultures [118]. Z-δ-viniferin (**4**) and Z-ε-viniferin (**5**) are reported from *V. vinifera* leaves following UV irradiation, while (+)-E-ε-viniferin (**6**) has been reported in *V. heyneana* stems and *V. vinifera* stems, as well as leaves [110].



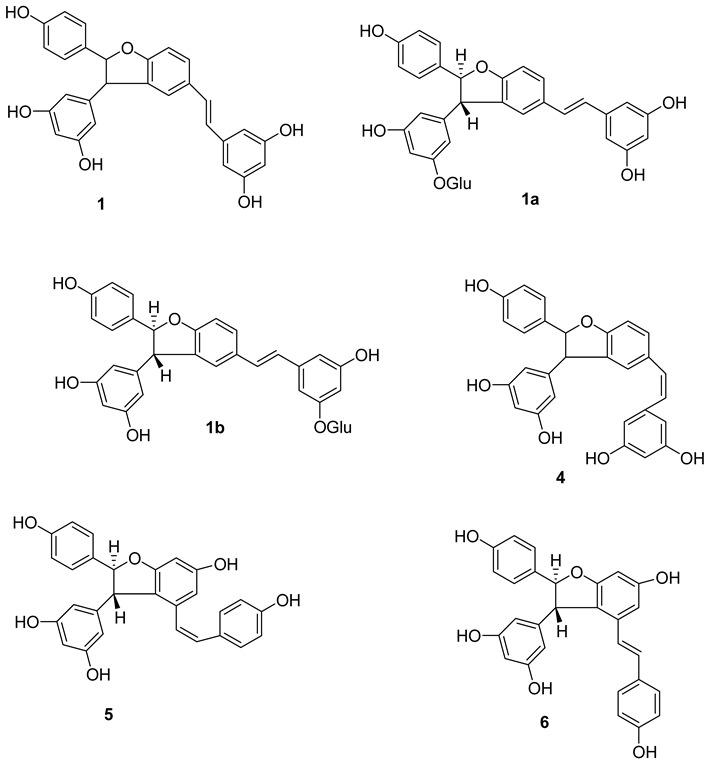



E-ε-viniferin (**7**) was reported in *V. vinifera* leaves, roots, stems and red wine [10]. Z-ω-viniferin (7a, 8a-*cis*-Z-£-viniferin) (**8**) and E-ω-viniferin (7a, 8a-*cis*-E-ε-viniferin) (**9**) were reported in *V. vinifera* leaves [110], whereas Z-ε-viniferin diglucoside (**10**) and E-ε-viniferin diglucoside (**11**) were reported in *V. vinifera* wine [119].



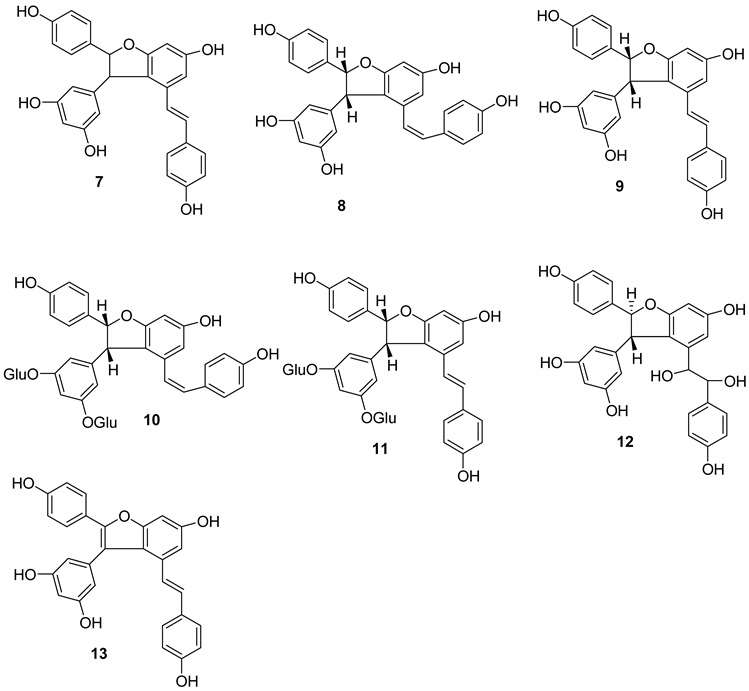



Furthermore, the presence of ε-viniferin diol (Betulifol B) (**12**) was reported in *V. betulifolia* stems [120] and Viniferifuran (**13**) (Amurensin H) was reported in *V. amurensis* roots [121].

Several research groups have focused on the synthesis of new resveratrol-derived chemical scaffolds with improved pharmacodynamics and pharmacokinetics with respect to the natural precursors. To develop synthetic procedures in order to investigate their biological activities, some methylated viniferins (**14–16**) were synthesized and characterized by spectral data [122].



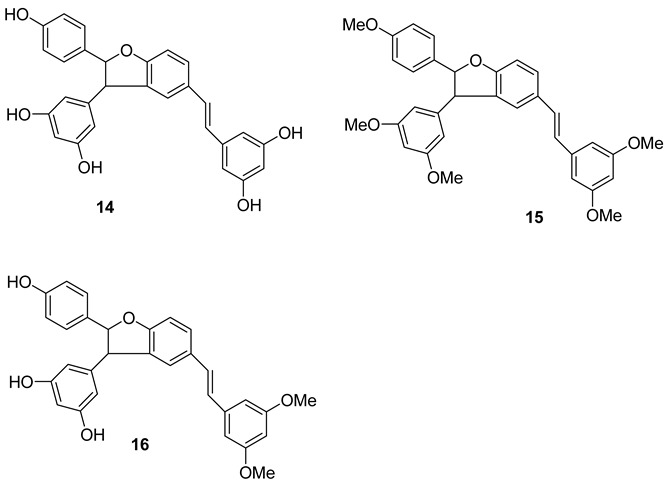



Recently, a collection of dehydro-viniferin analogues were synthesized and evaluated for their antimicrobial activities [107].



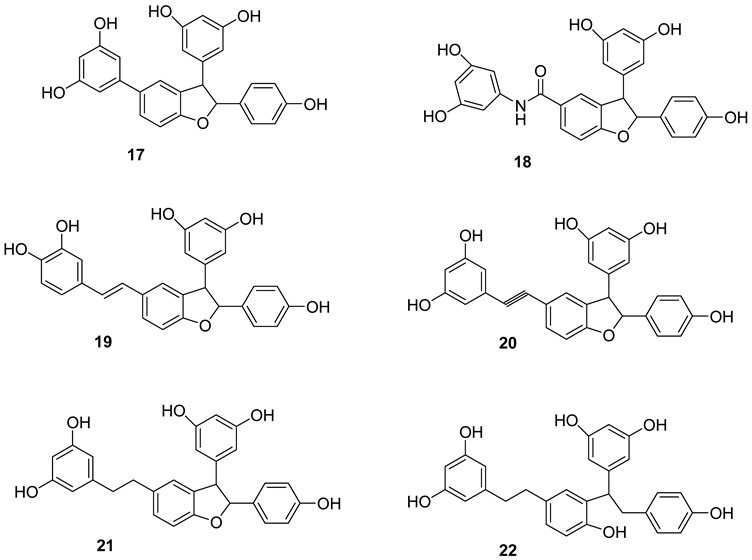



A chemical-structure analysis indicated that resveratrol was a polyphenol biphenyl, and that multiple hydroxyl groups affected its biological activities as well as *cis*- or *trans*-structures [123,124,125]. Resveratrol oligomers are characterized by the polymerization of two to eight resveratrol units and are the largest group of oligomeric stilbenes [126]. Resveratrol oligomer polyphenols were mainly isolated from five plant families, namely Vitaceae, Leguminosae, Gnetaceae, Dipterocarpaceae and Cyperaceae [126,127,128,129]. Nevertheless, although several studies showed various biochemical and pharmacological properties of resveratrol oligomers, so far, no systematic review has been conducted on these compounds. Their intricate structures and diverse biological activities are of significant interest for drug research and development and may provide promising prospects as cancer-preventive and therapeutic agents [84].

Resveratrol dimers: ε-viniferin (**A**) δ-viniferin (**B**), Heimiol A (**23**), Pallidol (**24**), Balanocarpol *α*-H (**25**), Ampelopsin β-H (**26**), Malibatol A (**27**) and Malibatol B (**28**). The phenol ε-viniferin, first isolated from *Vitis vinifera* (Vitaceae), is classified as a model for its biosynthesis from resveratrol [127]. Similar to resveratrol, ε-viniferin also attracted attention as a phytoalexin and was reported to have antifungal, antibacterial and antiviral activities. Furthermore, δ-viniferin, an isomer of ε-viniferin, only exists in plants in low concentrations.



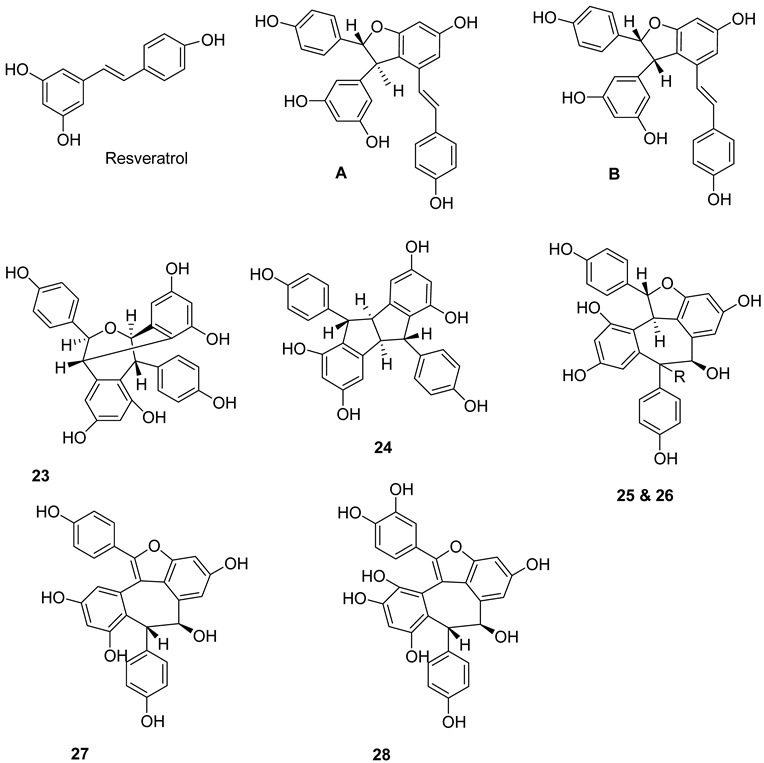



Resveratrol trimers: *α*-viniferin (**C**), miyabenol C (**29**), suffruticosol A *α*-H (**30**), suffruticosol B β-H (**31**) and gnetin H (**32**). Resveratrol trimers are formed by three resveratrol monomers through head-to-ligation or circular structure. The *α*-viniferin is a stilbene trimer isolated from *Caragana snice*, *Caragana chamlagu* and the stem bark of *Dryobalanops aromatica* [130].



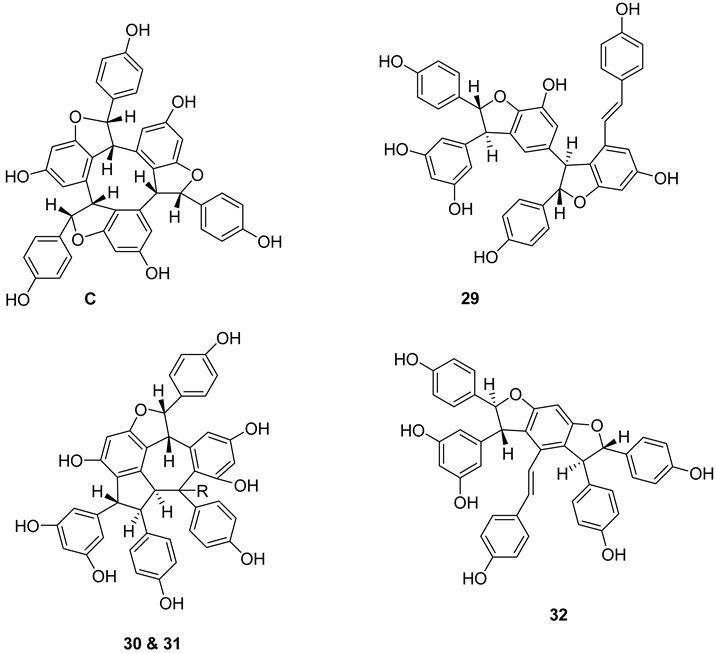



Resveratrol tetramers: Vaticanol C (**33**), kobophenol A (**34**) and hopeaphenol (**35**) from *Vitis vinifera*, a dimer, *trans*-*ε*-viniferin (**33**), as well as two tetramers, R2-viniferin (**34**) and r-viniferin (**35**) were obtained and evaluated for their cytotoxic activity to human hepatocellular carcinoma (HCC) cell lines p53 wild-type HepG2 and p53-null Hep3B. The distinctive toxicity of R2-viniferin on HepG2 was reported [82]. R-viniferin, also known as vitisin B [131] was found in a variety of grapevine-plant species [62].



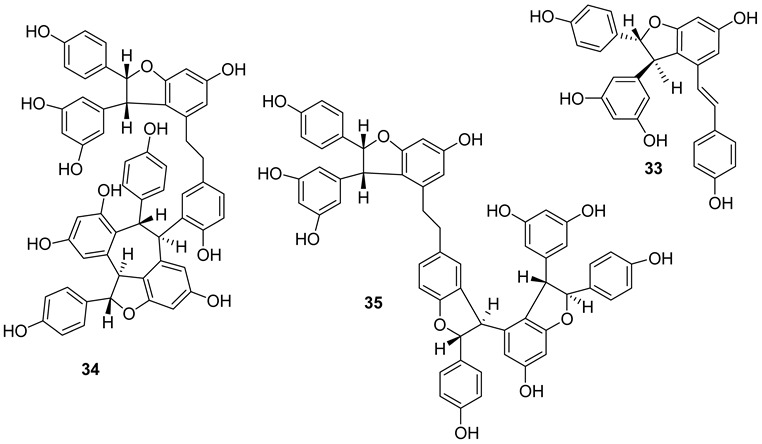



*trans*-*δ*-viniferin (**36**) was identified in downy-mildew-infected grapevine leaves and identified by HPLC coupled to mass spectrometry using atmospheric pressure photoionization (APPI–MSn) [132].



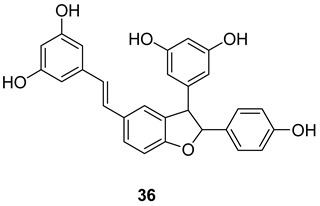



An acetyl cholinesterase inhibitor, γ-viniferin (**36a**), was reported in *Vitis vinifera* and a pharmaceutical composition made out of it was patented [133].



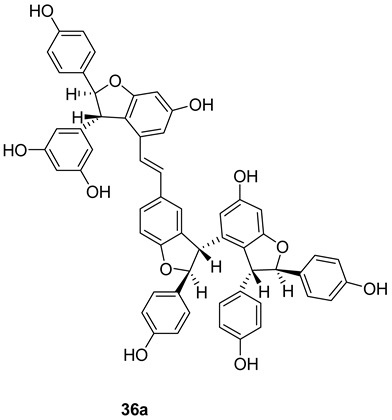



In the Structure **36b** and **36c** the hydroxyl groups are substituted with methyl and propyl groups and are called “cis viniferins”. When this is treated with 12M HCl in THF gives ampelopsin B [122]. 



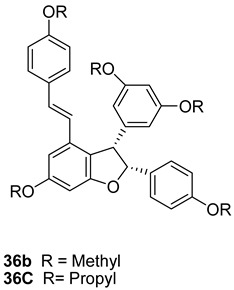



Overall, the focus of this review is on the chemistry and biological action of viniferin, as well as the development of consistent and ecologically friendly ways of commercializing the natural molecule on a large scale, since resveratrol has significant market potential. According to a survey in the Global Resveratrol Market Research Report (2020), the value of the global resveratrol market will reach USD 99.4 million by the end of 2026, as cited in Noviello et al. [114]. Only a small number of the viniferin-derived compounds mentioned above underwent pre-clinical research, which includes very few pharmacological studies. Researchers could forecast a few potential compounds in the near future, at least using in-silico studies; subsequently, they could work on them to develop molecules for clinical studies of various disorders. Future views should therefore emphasize the development of new therapeutics in relation to resveratrol-derived molecules, i.e., viniferins, which are capable of treating various diseases, as crucial sources of pharmaceuticals, as well as in other industries that can benefit humans. Therefore, further research may help to exploit its properties and its potential development into phyto-pharmaceuticals. This research will also have a significant impact on our understanding and provide the tools for novel and successful drug-discovery strategies.

## 6. Conclusions

The current review presents an all-inclusive literature search on various of viniferin studies through the years, especially on its anti-inflammatory, antipsoriasis, antidiabetic, antiplasmodic, anticancer, antiangiogenic, antioxidant, antimelanogenic, neurodegenerative effects, antiviral, antimicrobial, antifungal, antidiarrhea, anti-obesity and anthelminthic activities. The review shows the diverse collection of biological activities and possible applications in clinical research of all of the different forms of viniferin, such as *α*-viniferin, β-viniferin, δ-viniferin, ε-viniferin, γ-viniferin, vitisin A and B. Viniferins are resveratrol derivatives, one of the stilbenoids produced by plants as a defense mechanism in response to microbial attack, poisons, diseases, or UV-radiation. To mitigate the research, the confirmation of viniferin concentrations’ therapeutic efficacy in humans is still required. The pharmaceutical industry faces a significant challenge in applying this molecule clinically; it needs to be studied in greater depth to understand its bioavailability, metabolic pathways and human toxicity and, thus, the need to improve the field of clinical medicine is a challenge in producing commercially viable medicine. It may be quite cheap to produce large quantities while also being relatively safe, non-toxic, cost-effective and widely available. In the context of this review, viniferin has the potential to be used as a treatment for a wide range of human illnesses. Overall, viniferins are useful in medical technology and in the pharmaceutical, agricultural, nutraceutical and cosmeceutical sectors. We are confident that the information in this review will be more beneficial to researchers and industrial stakeholders working on the development of vinferin-based therapeutic medications.

## Data Availability

The data presented in this study are available on request from the corresponding author.

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
