# Peer review of "Chemistry, Biosynthesis and Pharmacology of Viniferin: Potential Resveratrol-Derived Molecules for New Drug Discovery, Development and Therapy"

_molecules, 2022, doi:10.3390/molecules27165072_

Round 1

Reviewer 1 Report

In the manuscript submitted, the authors present a review of the chemistry, biosynthesis, and pharmacology of viniferin.

Overall, the authors have provided a comprehensive and up-to-date overview of the literature on different forms of viniferin, its structural characterization, biosynthesis, and applications.

There are some minor issues that should be considered:

¾    The manuscript should be thoroughly checked for grammar and spelling mistakes,

¾    Figures containing molecular structures should be revised; there are numerous examples with distorted bond geometry (e.g. compound 20), five-membered ring, etc.

I consider this article suitable for publication in Molecules, after minor corrections.

Author Response

POINT BY POINT RESPONSE TO REVIEWER COMMENTS

Reviewer 1

In the manuscript submitted, the authors present a review of the chemistry, biosynthesis, and pharmacology of viniferin. Overall, the authors have provided a comprehensive and up-to-date overview of the literature on different forms of viniferin, its structural characterization, biosynthesis, and applications. There are some minor issues that should be considered:

Thank you so much for your comments and appreciation on the manuscript's structure. All of your suggestions have been taken into consideration in order to improve the quality of our manuscript.

No

Comments and Response

Page/Section/Location

1

¾   The manuscript should be thoroughly checked for grammar and spelling mistakes.   

      The English language and language flow has been rechecked and remodified appropriately in the revised manuscript. Thank you.

Complete Manuscript

2

Figures containing molecular structures should be revised; there are numerous examples with distorted bond geometry (e.g. compound 20), five-membered ring, etc. I consider this article suitable for publication in Molecules, after minor corrections.

In the revised manuscript, bond geometry has been examined and corrected where necessary for all chemical structures. Thank you.

Complete Manuscript

Reviewer 2 Report

The presented review is interesting but needs to be improved. All errors and sugestions are given below:

1.     Abstract – there is lack of numbering of mentioned compounds. First compound mentioned in line 40 viniferin must have number 1

2.     Generally, there is a huge mess in the numbering in whole manuscript number 1 belongs to veniferin line 103, amino acid line 192, resveratrol page 8, glycoside page 21 etc.

3.     The section 2.1 Distribution and sources of viniferin can not be accepted in current form. There need to be text describing the data in the Table 1. However almost nothing data are in that Table. More columns with additional informations is good to include at least 2 giving names of compound detected in mentioned plant, quantities of isolated/detected compounds.

4.     Instead of figure 1 only with names is good to give structures of forms of viniferin

5.     Page 6: in the section 2.3 once the Scheme (line 191) other time Figure (line 199) is used. Moreover the Scheme is good to give below this section not in section 2.4. Numbering in Scheme is not correct.

6.     All data like trans/cis/substitustion in aromatic ring and Latin names etc. are writing in italic for example lines: 194, 196, 197, 257, 275…

7.     Figure 4 not all mesomeric structures are presented

8.     Section 3 is not constructed well there is to much text not enough data. Instead of long parts with text important data can be given in table (giving studied model, doses, model of actions/mechanism etc)

9.     The most interesting part of presented topis is missing and only one sentence is given about search solutions for low bioavailability of viniferin and stilbenoids (ref. 68)

10.  Section 5 needs to be rewritten and completed

Author Response

POINT BY POINT RESPONSE TO REVIEWER COMMENTS

Reviewer 2

The presented review is interesting but needs to be improved. All errors and suggestions are given below:

Thank you so much for your comments and appreciation on the manuscript's structure. All of your suggestions have been taken into consideration in order to improve the quality of our manuscript.

No

Comments and Response

Page/Section/Location

1

Abstract – there is lack of numbering of mentioned compounds. First compound mentioned in line 40 viniferin must have number 1.

Thank you for your valuable suggestion. If numbering started at the beginning, all the starting materials and chemicals in biosynthesis would receive such continuation numbers, which would cause confusion with numbers mentioned in viniferin derivatives.

Hence, we have deleted all the numbers from Figure 2 and also in the biosynthesis part to prevent such kind of confusion. Instead, the chemical names are now provided in line with the chemical structures.

Figure 1 and section 2.3

2

Generally, there is a huge mess in the numbering in whole manuscript number 1 belongs to veniferin line 103, amino acid line 192, resveratrol page 8, glycoside page 21 etc.

Thanks for your insightful comment. To avoid this kind of misinterpretation, we removed the numbers from Figure 2 and the biosynthesis section. Instead, the chemical names are now provided in line with the chemical structures. Therefore, we started numbering beginning with section 5 (Compound 1-36). Thank you.

Figure 1 and section 2.3

3

The section 2.1 Distribution and sources of viniferin cannot be accepted in current form. There need to be text describing the data in the Table 1. However almost nothing data are in that Table. More columns with additional information’s is good to include at least 2 giving names of compound detected in mentioned plant, quantities of isolated/detected compounds.

I appreciate your suggestion. We have included a text that contain description of the Table 1. The type of viniferin isolated and reported in each plant has been included in Table 1. Thank you.

Table 1

4

Instead of figure 1 only with names is good to give structures of forms of viniferin.

Thank you for your suggestion. In the revised manuscript, the key structures are shown in Figure 1, and the chemistry and pharmacology sections explain for those compounds. The other types of viniferin structures is described in section 5 that are not been biologically tested much.

Figure 1 and 2, Section 5

5

Page 6: in the section 2.3 once the Scheme (line 191) other time Figure (line 199) is used. Moreover the Scheme is good to give below this section not in section 2.3. Numbering in Scheme is not correct.

I appreciate your suggestion. Schemes 1 and 2 have been replaced for Figures 3 and 4. The remaining figure numbers have all been adjusted in line with this changes. The numbering in the biosynthesis section (2.3) have been removed and mentioned the names in-line with the chemical structures. Thank you.

Scheme 1 and 2

6

All data like trans/cis/substitution in aromatic ring and Latin names etc. are writing in italic for example lines: 194, 196, 197, 257, 275…

I appreciate you pointing out this inconsistency in the manuscript. All trans/cis have been consistently italic throughout the entire revised manuscript.

Complete manuscript

7

Figure 4 not all mesomeric structures are presented.

Thank you for your feedback. We have included few mesomeric structures at the end (36a-c). Other than these two isomers to the best of my knowledge we couldn't find anything in the literature.

Page 28

8

Section 3 is not constructed well there is too much text not enough data. Instead of long parts with text important data can be given in table (giving studied model, doses, model of actions/mechanism etc).

Thank you very much for your feedback. Figure 4, 5 and 6 in the revised manuscript explained it’s mechanism of action. If we convert into Table form, it looks a kind of repetition. However, we have gone through the section 3 and made it concise form and removed certain information which are lengthy. Thank you.

Section 3, Figure 4, 5 and 6

9

The most interesting part of presented topic is missing and only one sentence is given about search solutions for low bioavailability of viniferin and stilbenoids (ref. 68).

I appreciate your input. The final paragraph of the bioavailability section explains the difficulties and opportunities for increasing viniferin's bioavailability (2.4). Thank you.

Section 2.4

10

Section 5 needs to be rewritten and completed.

The final paragraph of Section 5 has been revised, and we are provided some opportunities to get beyond all the viniferin molecules' drawbacks and turn them into drugs in the near future. Thank you.

Section 5

Reviewer 3 Report

This work is devoted to a literary review of the current state of research in the field of resveratol and viniferin. There are several forms of viniferin, each of which has biological activity and can be used as a drug in the future.

In general, the review is informative, structured and of value both to researchers in this field and to a wide range of readers.

However, there are a number of shortcomings and errors:

1) Line 82-93. Information is redundant. There are 132 sources in the list of references, among which there are articles of 1977 and 1982.

2) It is desirable to further emphasize the importance and direction of the review

3) Table 1. Really? Maybe it is a picture? The text should contain a description of the table

4) For many biologically active substances of natural origin, their content in the source is important. Maybe it's worth adding a column with information about the content of viniferin to Table 1?

5) £-viniferin (£ is the pound symbol)

6) "Trimmer" is incorrect. Trimer - a molecule consisting of three units of monomers.

7) line 128 Reference to UV spectroscopy data is required.

8) line 145 formulae

9) 1H-1H should indicate that this is COSY (as in the source)

10) line 166-167 reference to table 1 is incorrect

11) Table 2 - citation required

12) line 636. unification of designations is needed (E-viniferin, and Z-viniferin)

13) Section 5 is an important part of the review, but its placement in the text is not entirely logical. Maybe the authors should rephrase and move section 5 to the beginning of the review?

14) line 814 "#78@@author-year}." ?

15) Too many authors.

Author Response

POINT BY POINT RESPONSE TO REVIEWER COMMENTS

Reviewer 3

This work is devoted to a literary review of the current state of research in the field of resveratol and viniferin. There are several forms of viniferin, each of which has biological activity and can be used as a drug in the future. In general, the review is informative, structured and of value both to researchers in this field and to a wide range of readers. However, there are a number of shortcomings and errors:

Thank you very much for your insightful feedback. We have taken into account all of your comments in order to improve the quality of our manuscript.

No

Comments and Response

Page/Section/Location

1

Line 82-93. Information is redundant. There are 132 sources in the list of references, among which there are articles of 1977 and 1982.

In the revised manuscript, the above-mentioned sentence has been altered appropriately. Thank you.

Page 3, Introduction

2

It is desirable to further emphasize the importance and direction of the review

Thank you for the suggestion. In the closing paragraph of the introduction section, we discussed about the importance and scope of the review.

Introduction

3

Table 1. Really? Maybe it is a picture? The text should contain a description of the table.

I appreciate your suggestion. We have included a text that contain description of the Table 1.

Page 3, Table  1

4

For many biologically active substances of natural origin, their content in the source is important. Maybe it's worth adding a column with information about the content of viniferin to Table 1?

I appreciate your comments. . The type of viniferin isolated and reported in each plant has been included in Table 1. We believe that Table 1 is more suited to describe such information in depth because it makes it simple for readers to identify the specific plant and its parts that contain viniferin as well as the reference of that specific information.

Table 1

5

£-viniferin (£ is the pound symbol)

I appreciate you noticing this mistake. In the revised manuscript, we have changed “£-viniferin” to “ε-viniferin”. Thank you.

Complete manuscript

6

"Trimmer" is incorrect. Trimer - a molecule consisting of three units of monomers.

Thank you for spotting this error. In the revised manuscript, we have changed "Trimmer" to "Trimer".

Page 4 and 25

7

line 128 Reference to UV spectroscopy data is required.

The reference for UV spectroscopy data has been included in the revised manuscript. Thank you.

Page 4

8

line 145 formulae

Thank you for spotting this error. In the revised manuscript, we have changed “formulae” to “formula”. Thank you.

Page 5

9

1H-1H should indicate that this is COSY (as in the source)

I appreciate you pointing this out. In the revised manuscript, we have changed “1H-1H NMR” to “1H-1H-COSY NMR”. Thank you.

Page 5

10

 line 166-167 reference to table 1 is incorrect

Thank you again for pointing out this error. Yes, Table 1 citation in the main body text has been replaced with Table 2 since only Table 2's content is relevant.

Page 5

11

Table 2 - citation required

Table 2 was cited in the main body text at the relevant place. Thank you.

Page 5

12

line 636. unification of designations is needed (E-viniferin, and Z-viniferin)

I appreciate you bringing out this mistake. The word "E-viniferin" has been altered to appear consistently as "E-ε-viniferin" throughout the manuscript. Thank you.

Page 18

13

Section 5 is an important part of the review, but its placement in the text is not entirely logical. Maybe the authors should rephrase and move section 5 to the beginning of the review?

I appreciate your comment. Section 5 has been rewritten and given a better title. This section's placement has not change. Since the flow of this review is to explore viniferin's chemistry and pharmacology first, then the importance its derivatives in the development and discovery of new drugs. We hope the reviewer will accept our justification for including this part at the end of the review. Thank you.

Section 5

14

line 814 "#78@@author-year}." ?

I appreciate you pointing out this error. The aforementioned error has been fixed in the revised version of the manuscript. Thank you.

Page 26

15

Too many authors.

This review has progressed from the contributions of all authors. Details were provided in the section on author contributions. Thank you.

Author contributions section

Round 2

Reviewer 3 Report

The authors corrected the article according to the comments.

The article may be published in Molecules journal.